# Maximizing the Therapeutic Effect of Endothelin Receptor Antagonists in Pulmonary Fibrosis: A Paradigm for Treating the Disease

**DOI:** 10.3390/ijms25084184

**Published:** 2024-04-10

**Authors:** Jerome Cantor

**Affiliations:** School of Pharmacy and Health Sciences, Queens, NY 11439, USA; jocantor1@gmail.com

**Keywords:** pulmonary fibrosis, endothelin, endothelin receptor antagonists, bleomycin, amiodarone, lipopolysaccharide, emergent phenomena

## Abstract

Using a lipopolysaccharide model of acute lung injury, we previously showed that endothelin-1 (ET-1), a potent mediator of vasoconstriction, may act as a “gatekeeper” for the influx of inflammatory cells into the lung. These studies provided a rationale for testing the effect of HJP272, an endothelin receptor antagonist (ERA), in hamster models of pulmonary fibrosis induced by intratracheal instillation of either bleomycin (BLM) or amiodarone (AM). To determine the temporal effects of blocking ET-1 activity, animals were given HJP272 either 1 h before initiation of lung injury or 24 h afterward. The results indicated that pretreatment with this agent caused significant reductions in various inflammatory parameters, whereas post-treatment was ineffective. This finding suggests that ERAs are only effective at a very early stage of pulmonary fibrosis and explains their lack of success in clinical trials involving patients with this disease. Nevertheless, ERAs could serve as prophylactic agents when combined with drugs that may induce pulmonary fibrosis. Furthermore, developing a biomarker for the initial changes in the lung extracellular matrix could increase the efficacy of ERAs and other therapeutic agents in preventing the progression of the disease. While no such biomarker currently exists, we propose the ratio of free to peptide-bound desmosine, a unique crosslink of elastin, as a potential candidate for detecting the earliest modifications in lung microarchitecture associated with pulmonary fibrosis.

## 1. Introduction

The objective of the current paper is to provide a framework for understanding how the pathogenesis of pulmonary fibrosis is related to the potential efficacy of treatments for the disease. Based on a series of experiments involving animal models of pulmonary fibrosis, it was hypothesized that the progression of the disease is programmed at an early stage of development. Effective treatment may therefore require the identification of biomarkers that can detect the initial morphological and biochemical changes in the lung parenchyma and permit timely therapeutic intervention. This approach is supported by the concept of emergent phenomena, which suggests that biochemical and morphological alterations associated with the fibrotic process may rapidly reach a critical threshold involving a phase transition where the disease becomes less amenable to treatment.

Pulmonary fibrosis is a common feature of interstitial lung disease and is characterized by a complex reorganization of the lung architecture that includes changes in the structure and cellular composition of small airways and alveolar walls. This remodeling depends on the activity of inflammatory mediators that regulate leukocyte recruitment, alveolar epithelial hyperplasia, and extracellular matrix deposition [1]. The initiation and progression of this process may involve endothelins, which comprise a group of homologous peptides with potent vasoconstrictive properties [2,3,4]. This is supported by the main isoform, endothelin-1 (ET-1), which is a 21-amino acid peptide formed by the enzymatic cleavage of a much larger precursor, prepro ET-1, to big ET-1, which is then converted to ET-1.

This laboratory has previously shown that ET-1 may function as an inflammatory cell “gatekeeper,” facilitating the migration of neutrophils from the vascular compartment to the lung. Treatment of lipopolysaccharide (LPS)-induced acute lung injury with HJP272, a noncommercial ET-1 receptor antagonist (ERA), significantly reduced multiple inflammatory parameters [5,6,7]. These included lung morphological changes, neutrophil levels in bronchoalveolar lavage fluid (BALF), and tumor necrosis factor receptor 1 (TNFR1) expression by BALF macrophages. In contrast, exogenous administration of ET-1 had the opposite effect, significantly increasing the amount of BALF neutrophils following short-term exposure to cigarette smoke [7].

The effect of ET-1 on neutrophils may involve increased synthesis of p-selectin and intercellular adhesion molecule 1, which facilitates the attachment of these cells to vascular endothelium [8]. ET-1 has also been shown to increase neutrophil expression of CXCR2, which binds interleukin-8, a potent activator of these cells [9]. Furthermore, ET-1-induced changes in the F-actin content of neutrophils could promote their sequestration in pulmonary microvessels [10].

The significant reduction in BALF neutrophils after treatment with HJP272 suggests that the anti-inflammatory activity of this agent involves blocking the effects of ET-1 on the vascular compartment. Nevertheless, other factors may also contribute to this finding, including a decrease in the expression of macrophage TNFR1, which could limit TNF-alpha-induced synthesis of metalloproteinases that recruit inflammatory cells to the lung [11,12].

Based on the anti-inflammatory activity of HJP272 in LPS-induced lung injury, the effect of this agent was studied in experimental models of pulmonary fibrosis induced by bleomycin (BLM) or amiodarone (AM), drugs whose side effects include fibrotic lung injury. Intratracheal instillation of BLM or AM in hamsters produces morphological changes within several weeks that are similar to those seen in human pulmonary fibrosis.

The effects of HJP272 on pulmonary fibrosis were determined by giving animals a single dose of this agent either one hour before induction of lung injury or 24 h afterward. The study was designed to test the hypothesis that ET-1 plays an important role at the earliest stage of the disease by regulating the influx of neutrophils into the lung. If this proposition is correct, then post-treatment with HJP272 should be less effective in preventing subsequent inflammation and fibrosis. While the use of ERAs in the BLM model has previously been shown to reduce pulmonary fibrosis, the time-dependent effects have not been investigated [13,14].

## 2. ERAs in Animal Models of Pulmonary Fibrosis

### 2.1. BLM-Induced Pulmonary Fibrosis

The BLM model is commonly used to study the pathogenesis of interstitial pulmonary fibrosis because it has morphological features that are similar to the human disease, including a marked influx of inflammatory cells, alveolar epithelial hyperplasia, airway distention, and interstitial fibrosis (Figure 1). The mechanism of lung injury involves the formation of complexes between this agent and Fe^2+^, which generate free radicals that damage DNA [15].

The rapid development of inflammation and fibrosis following treatment with BLM suggests that this form of lung injury may be better characterized as a wound-healing phenomenon rather than lung remodeling. The changes produced by BLM cannot duplicate the gradual progression of the human disease, and cessation of treatment may reverse previous morphological changes. Nevertheless, this model has provided a better understanding of the mechanisms that may be responsible for the development of human pulmonary fibrosis.

Although the LPS and BLM models have different pathogenetic mechanisms, a rapid influx of neutrophils is seen following the instillation of either agent [16,17]. In the LPS model, these cells quickly decrease, and the lung shows few residual morphological changes [8]. However, repeated exposure to LPS can lead to pulmonary emphysema due to these cells’ continued release of elastases [18].

In contrast, a single instillation of BLM causes progressive inflammatory changes that culminate in marked lung remodeling. This process takes several weeks, with the fibrotic lesions becoming most pronounced within the first month. The rapid development of fibrosis provides an opportunity to investigate the effects of ERAs over the entire course of the disease. By varying the temporal relationship between ERA treatment and BLM instillation, it might be possible to determine where ET-1 exerts the greatest effect on the pathogenesis of pulmonary fibrosis.

To test this concept, hamsters were given an intraperitoneal injection of HJP272 one hour before intratracheal instillation of BLM or 24 h afterward [19]. Lung injury was assessed by measuring the following parameters at 2 to 4 weeks post-BLM: lung histopathological changes, neutrophil content in bronchoalveolar lavage fluid (BALF), lung collagen content, tumor necrosis factor receptor 1 (TNFR1) 1 expression by BALF macrophages, and alveolar septal cell apoptosis [19]. During this period, additional studies determined BALF levels of transforming growth factor (TGF) beta-1, stromal cell-derived factor 1 (CXCL12), and platelet-derived growth factor-BB.

For all of these variables, pretreatment with HJP272 caused significant decreases compared to those receiving BLM alone, whereas post-treatment was ineffective in reducing their levels [19]. The differences between the treatment groups were most evident morphologically, where the lungs of animals pretreated with HJP272 showed much less fibrosis, suggesting that the initial inflammatory events determine the extent of lung injury (Figure 2). The level of disease in each group was scored and expressed as a “fibrotic index” (Figure 3) [20].

These findings are consistent with clinical trials showing that ERAs are ineffective in treating human pulmonary fibrosis when administered after the disease becomes clinically apparent. Nevertheless, they support the hypothesis that they may serve as prophylactic agents when given in combination with drugs that have fibrogenic potential.

### 2.2. AM-Induced Pulmonary Fibrosis

Microscopic examination of hamster lungs at three weeks following intratracheal instillation of AM revealed inflammatory cell infiltrates, airspace remodeling, and interstitial fibrosis (Figure 4). While it was originally proposed that intracellular accumulation of phospholipids may play a critical role in the disease process, a subsequent study comparing the pulmonary response in hamsters to either oral or intratracheal treatment with AM showed that phospholipidosis is not a critical mechanism in the development of pulmonary fibrosis, suggesting that other mechanisms are in AM-induced lung fibrosis [21,22].

The increased levels of reactive oxygen species after treatment with AM suggests that injury to cell and organelle membranes may play an important role in the pathogenesis of the disease (Figure 5) [23,24]. Furthermore, eosinophils in the lungs of patients receiving AM indicate the possible coexistence of an immune-mediated mechanism [25].

As with the BLM model, HJP272 was administered intraperitoneally 1 h before treatment with AM or 24 h afterward [26]. The parameters of inflammation examined at 2 to 4 weeks post-AM were the same as those used to evaluate BLM-induced pulmonary fibrosis, and the results were similar to those seen with that model. Pretreatment with HJP272 produced significant reductions in all of these variables compared to AM alone, while post-treatment was ineffective, providing additional support for the hypothesis that the course of injury and repair is programmed at a very early stage of the inflammatory process [26]. The temporal effects of HJP272 were again most evident by examining lung morphological changes, which showed a much more limited fibrotic reaction in the pretreatment group (Figure 6). As with BLM-induced fibrosis, the morphological changes were expressed in terms of a “fibrotic index” (Figure 7).

The findings from the BLM and AM models are consistent with the hypothesis that the initial events in the development of pulmonary fibrosis involve common mechanisms of injury and that ERAs may, therefore, have broad efficacy as prophylactic agents when used in combination with drugs that have fibrogenic potential.

### 2.3. ERA Modulation of Fibrosis in the BLM and AM Models

Although the reduction in inflammation and fibrosis following pretreatment with HJP272 may result from blocking the activity of ET-1 in the vascular compartment, other factors may contribute to these findings. The reduced expression of macrophage TNFR1 may lower TNF-alpha-induced migration of leukocytes to the lung [10,11,12]. Furthermore, the diminished levels of PDGF, TGF-β, and CXCL12 may impair fibrogenesis by limiting the influx of fibrocytes into the lung [27,28,29,30,31,32,33]. ET-1 may also contribute to the differentiation of fibrocytes by inducing the expression of connective tissue growth factor [34].

Whether other ERAs have the same effect on fibrogenesis may depend on their specific binding properties. HJP272 is a selective ERA with a primary affinity for the ET-1 subtype A receptor, whereas mixed antagonists also bind to ET-1 subtype B receptors. One study showed that repeated administration of the Bosentan, a mixed ERA, significantly decreased BLM-induced pulmonary fibrosis. However, the effect of a single dose at different time points was not determined [14].

## 3. Therapeutic Considerations

### 3.1. The Potential Role of Disease Emergence in Pulmonary Fibrosis

Clinical trials using both types of ERAs have been unsuccessful in preventing the progression of pulmonary fibrosis. However, a subset of patients with milder disease showed a trend toward improved survival with Bosentan [35,36,37,38]. The limited efficacy of ERAs may be related to the phenomenon of emergence, where complex interactions at different levels of scale produce a spontaneous reorganization of the lung involving the remodeling of the pulmonary architecture [39].

The process of emergence is a common feature of complex systems such as chemical reactions, epidemics, and disease pathogenesis. It may be represented by percolation models based on the random movement of fluids through interconnecting channels [40]. The convergence of isolated currents in the network eventually reaches a critical threshold involving a phase transition comprising a change in the structure and behavior of the system. In the case of pulmonary fibrosis, the spread of the extracellular matrix through the lung interstitium is similar to the diffusion of fluid through a percolation network. It produces analogous changes in the chemical and physical properties of the lung.

The increased collagen deposition as part of this process alters the elastic modulus of the alveolar walls and modifies the transmission of mechanical forces related to breathing. This may lead to further interstitial injury and repair, resulting in a self-propagating extension of the disease on a much larger scale. Computer-generated percolation models of pulmonary fibrosis support the validity of this mechanism by showing that local changes in alveolar wall structure evolve into global morphological alterations that resemble those seen in this disease [41,42].

### 3.2. Developing a More Effective Treatment for Pulmonary Fibrosis

The differential effects of HJP272 before and after initiating lung injury provide a rationale for developing drugs that target the broader process of disease emergence. The complexity of the events associated with phase transitions suggests that the loss of a specific molecular component of the inflammatory reaction, such as ET-1, could be circumvented by higher-level structural alterations in the alveolar walls.

In addition to explaining why ERAs are largely ineffective in preventing the progression of pulmonary fibrosis, the concept of emergence emphasizes the need for developing biomarkers with the sensitivity and specificity to detect and treat the disease before it becomes less amenable to therapeutic intervention [43]. The availability of a biomarker with these properties might have the additional effect of enhancing the role of ERAs in treating pulmonary fibrosis.

## 4. Elastin Crosslinks as a Potential Biomarker for Pulmonary Fibrosis

### 4.1. The Ratio of Peptide-Free to Peptide-Bound Elastin Crosslinks

The core elastin protein of elastic fibers, which is responsible for their distensibility, undergoes marked injury and repair in pulmonary fibrosis [44]. Due to the very low turnover of this matrix component in healthy lungs, their breakdown products may serve as a biomarker for this disease. In particular, the level of the elastin-specific crosslinking amino acids, desmosine and isodesmosine (DID), have been used to monitor a number of lung diseases that involve changes in the structure and composition of the extracellular matrix. In the case of pulmonary fibrosis, various studies have documented increases in DID in both the BLM model of pulmonary fibrosis and patients with this disease [44,45].

Previously, our laboratory reported a substantial positive correlation between DID and the degree of alveolar wall injury in models of pulmonary fibrosis and emphysema induced by intratracheal BLM and elastase, respectively [46]. While both forms of lung injury involved increases in BALF DID, the proportion of peptide-free crosslinks was very different in the two models. In elastase-induced pulmonary emphysema, the ratio of free to peptide-bound DID was markedly increased, whereas this parameter was decreased in the BLM model of pulmonary fibrosis [45,46].

Based on these findings, it is proposed that this ratio may be used to distinguish between the microarchitectural changes that are present in the two diseases. Further support for this hypothesis was seen in a 28-day clinical trial involving aerosolized hyaluronan as a treatment for pulmonary emphysema [47]. In that study, the ratio of free/bound DID was significantly reduced (*p* < 0.05) over the 35-day measurement interval in patients treated with hyaluronan whereas the placebo showed no effect, suggesting the feasibility of using this biomarker to measure changes in the lung extracellular matrix (unpublished results). Earlier published studies showed a significant correlation between an increase in free to bound DID in urine and loss of lung function in human pulmonary emphysema, as measured by forced expiratory volume at one second [48]. However, it remains to be determined whether treatment with ERAs alters the free to peptide-bound DID ratio in pulmonary fibrosis, so the role of this putative biomarker in detecting a therapeutic effect remains speculative.

The proportional decrease in free DID in BLM-induced fibrosis may be due to the increased preservation of intact elastic fibers in pulmonary fibrosis. The surrounding extracellular matrix may act as a structural support and a protective barrier against enzymatic injury of these fibers. In particular, an increase in hyaluronan content, which binds to elastic fibers, may limit the effects of elastases and other injurious agents on elastin [48].

### 4.2. Modeling the Effects of Pulmonary Fibrosis on DID Crosslinking

To better understand the relationship between DID and pulmonary fibrosis, we modeled the effects of altered mechanical forces on patterns of elastic fiber breakdown. Here again, percolation models provide a useful means of analyzing the effects of these forces on elastic fibers. The particular model used for this purpose involves a network composed of normal (K1) and rigid (K2) alveolar walls [49]. Under these conditions, the dispersion of mechanical forces through the lattice depends on the K1 to K2 ratio (Figure 8).

The properties of percolation networks suggest that the ratio of K1 to K2 may denote a larger pattern of change involving the structure of the extracellular matrix. At this lower level of scale, alveolar walls with normal amounts of collagen correspond to K1 units, while the K2 component reflects walls with increased collagen. The uneven mechanical forces caused by increased K2 units would render the normal regions more susceptible to dilatation and rupture, resulting in the fragmentation of elastic fibers and the loss of DID crosslinks. However, as collagen deposition encompasses larger areas of the lung, the level of free DID would decrease in response to the reduced effects of mechanical stress on the increasing number of rigid alveolar walls, resulting in a lower ratio of free to peptide-bound DID.

Another component of the fibrotic reaction that may affect the free to peptide-bound DID ratio is lysyl oxidase, an enzyme responsible for the formation of DID crosslinks from lysine residues in elastin peptides. While increased levels of lysyl oxidase were seen in the lungs of rats treated with BLM, a similar finding was noted in hamsters treated with elastase [50,51]. The lack of correlation between lysyl oxidase levels and morphological changes suggests that the free-to-bound DID ratio may not depend upon lysyl oxidase activity.

### 4.3. Potential Limitations of the DID Biomarker

Numerous inflammatory mediators have been proposed as potentially useful biomarkers for pulmonary fibrosis but may not reflect specific lung microarchitectural changes. If percolation processes are active in the lung, then a proportionate decrease in the free to peptide-bound DID ratio may serve as a sensitive and specific indicator of early structural changes in elastic fibers that predispose the lung to developing pulmonary fibrosis.

The relationship between connective tissue crosslinking and lung structural changes was shown in a study involving the use of a lysyl oxidase inhibitor, beta-aminopropionitrile (BAPN), in a cadmium chloride model of pulmonary fibrosis. Animals receiving BAPN had morphological changes consistent with pulmonary emphysema, whereas those left untreated developed pulmonary fibrosis [52].

Nevertheless, the DID levels in blood or urine levels may lack specificity as a biomarker of lung injury due to the release of DID crosslinks from sites other than the lung, such as blood vessels and joints. The co-existence of diseases such as arteriosclerosis or osteoarthritis, which also involve elastic fiber damage, may obscure increases in DID levels due to pulmonary emphysema. Studies suggest that the level of DID in urine may be more closely related to cardiovascular disease than to pulmonary function, and various factors such as age, sex, body mass index, and smoking habits may also impinge upon the sensitivity and specificity of this biomarker for alveolar wall damage [53,54].

Despite these limitations, an advantage of the free to peptide-bound DID ratio is the absence of a dimensional unit, which makes it independent of the negative effects of large variations in absolute values. Furthermore, the use of sputum and possibly breath condensate might increase the specificity of DID for lung elastic fiber injury. Nevertheless, the acceptance of free DID as a biomarker of pulmonary emphysema may ultimately depend on developing an accurate and reproducible method of analysis. Currently, mass spectrometry provides the best means of measuring DID, but the high cost of the equipment may be a rate-limiting factor in the widespread adoption of the biomarker [55].

### 4.4. Future Directions of Biomarker Research

In silico percolation models have been used to investigate seemingly unrelated molecular and macroscopic events, particularly when they are associated with emergent phenomena. While the application of these models to pulmonary fibrosis is still at an early stage, they have already become a valuable tool in identifying potential biomarkers that more accurately mirror the specific dynamics of the disease.

Initial studies employed simple models to determine the effect of mechanical forces on the lung. A network of springs was modified by either increasing their stiffness or removing them entirely, resulting in architectural changes similar to those seen in the lungs of patients with pulmonary fibrosis and emphysema, respectively [41]. These changes did not produce significant alterations in lung structure until a critical threshold was reached, suggesting the emergence of lung disease may be predicted by examining early microscopic and molecular phenomena.

Regarding pulmonary fibrosis, inflammatory mediators may be less reliable biomarkers of the disease than structural molecules such as DID because they are unrelated to the mechanical forces involved in alveolar wall injury. As a result, they may lack specificity for the disease and, therefore, have less prognostic value. In addition to DID, structural biomarkers could include the breakdown products of collagen, where the ratio of type 1 to type 3 collagen might increase as the fibrotic reaction progresses. The change in the proportion of collagen subtypes could be combined with free/bound DID and other connective tissue modifications to form a panel of biomarkers that reflect early events in the emergence of pulmonary fibrosis.

The accuracy and reproducibility of these biomarker measurements may be enhanced by using mass spectrometry, which allows more precise characterization of the amount and molecular weight of extracellular matrix components that may serve as specific indicators of pulmonary fibrosis. The levels of these molecules could be correlated with widely accepted parameters of disease progression, thereby providing surrogate endpoints for clinical trials of novel agents for treating pulmonary fibrosis. The availability of a sensitive and specific molecular biomarker may circumvent the prolonged interval needed for physiological or radiographic determination of therapeutic efficacy. This would decrease the resistance to testing novel treatment agents and expedite the development of more effective treatments for pulmonary fibrosis.

## 5. Conclusions

Pulmonary fibrosis involves multiple interactions at different levels of scale that produce marked architectural changes, including alveolar epithelial hyperplasia, distention of airspaces, and interstitial fibrosis. The temporally disparate results associated with HJP272 treatment in the BLM and AM models suggest that therapeutic intervention may be most effective at an early stage of the disease, prior to the development of widespread biochemical and morphological changes. The ability to detect pulmonary fibrosis at an early stage will depend on a better understanding of the molecular and macroscopic patterns of behavior that reflect the emergence of the fibrotic reaction. In silico modeling of the disease may facilitate the identification of molecular processes that reflect the initial proliferation of cellular and connective tissue components before their convergence into clinically apparent disease. The availability of a biomarker for the emergent properties of pulmonary fibrosis may permit more timely intervention with therapeutic agents, including ERAs, that delay the worst features of the disease and prevent respiratory failure.

## Figures and Tables

**Figure 1 ijms-25-04184-f001:**
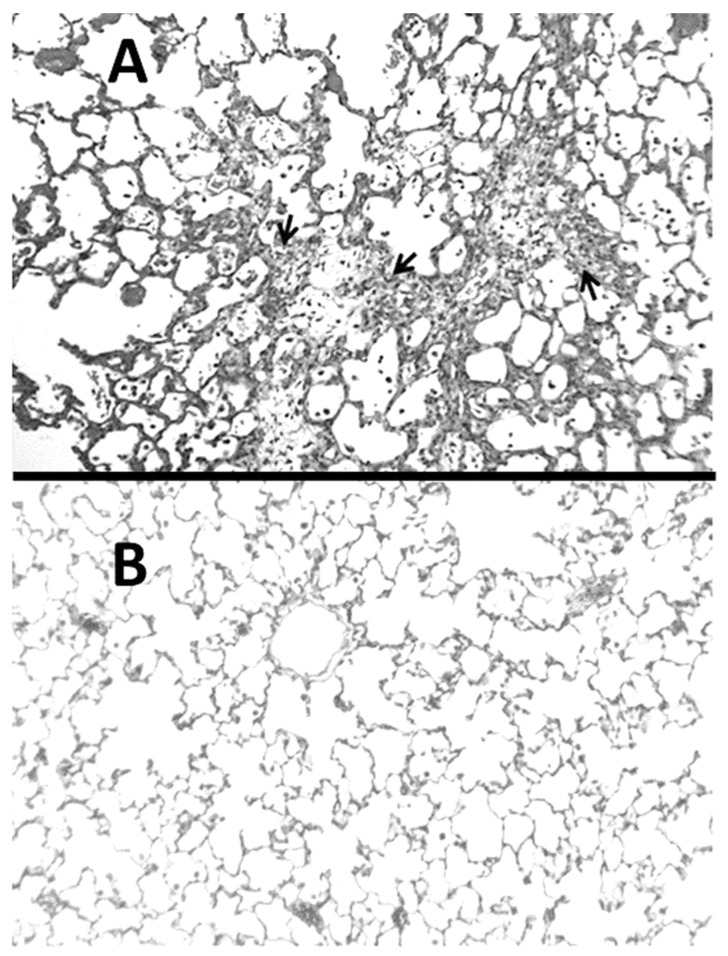
(**A**) Hamster lung at three weeks post-instillation of BLM showed inflammatory cell infiltrates and marked interstitial thickening with fibrosis (indicated by arrows). (**B**) Normal lung for comparison.

**Figure 2 ijms-25-04184-f002:**
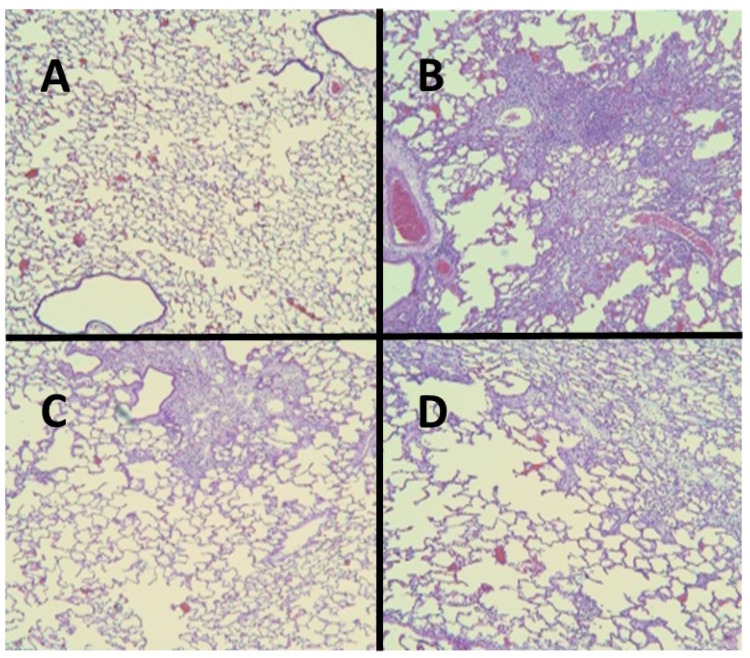
(**A**) Normal hamster lung. (**B**) Intratracheal instillation of BLM alone induced extensive pulmonary fibrosis. (**C**) Treatment with HJP272 before BLM significantly decreased fibrosis. (**D**) Treatment with HJP272 after BLM was much less effective in reducing fibrosis.

**Figure 3 ijms-25-04184-f003:**
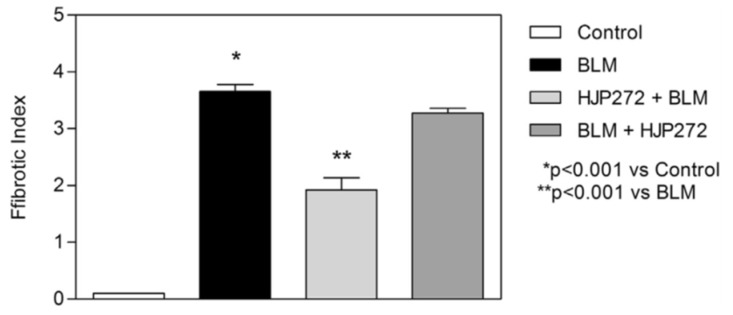
Graph showing the level of BLM-induced fibrosis in the various treatment groups, as measured by the fibrotic index.

**Figure 4 ijms-25-04184-f004:**
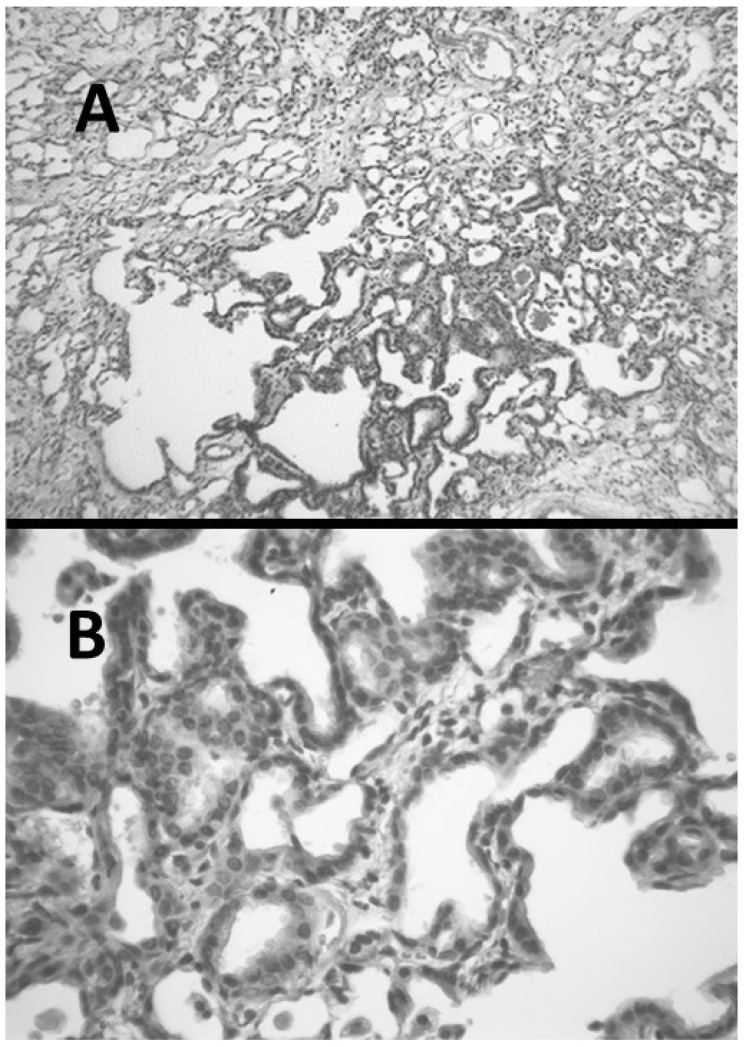
(**A**) Hamster lung at three weeks post-instillation of AM showing inflammation and marked interstitial fibrosis. (**B**) Lung with interstitial fibrosis and alveolar epithelial hyperplasia.

**Figure 5 ijms-25-04184-f005:**
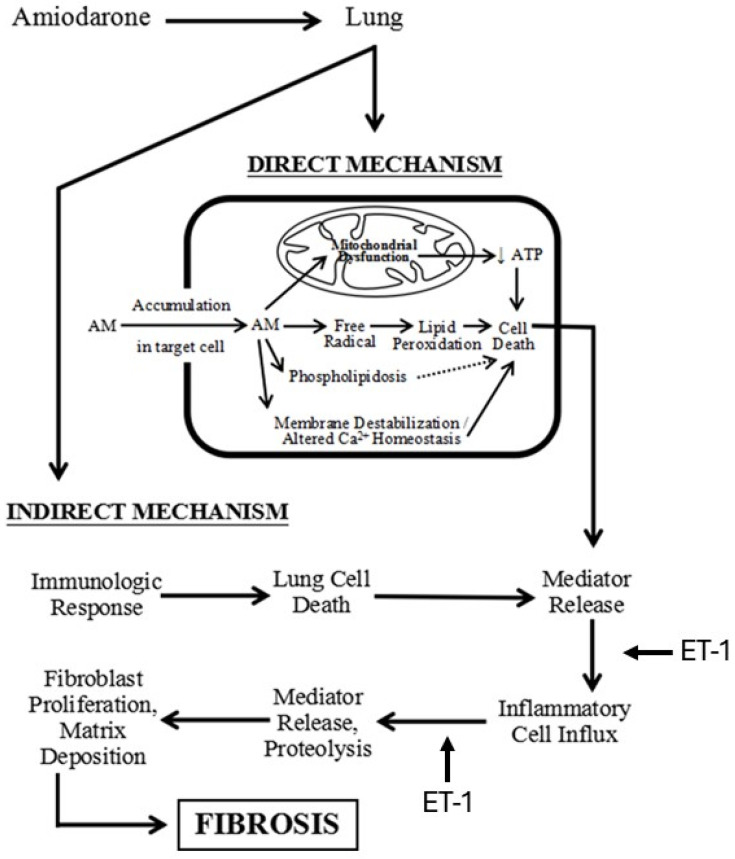
Diagram showing the mechanism of AM-induced pulmonary fibrosis and the pathways where ET-1 may play a role in the fibrotic process.

**Figure 6 ijms-25-04184-f006:**
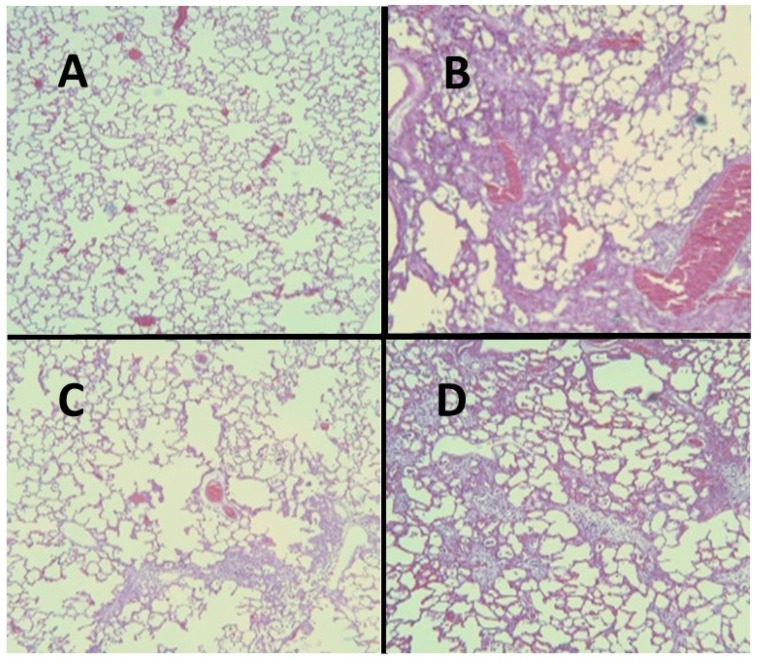
(**A**) Normal hamster lung. (**B**) Intratracheal instillation of AM alone induced extensive pulmonary fibrosis. (**C**) Treatment with HJP272 before AM markedly reduced fibrosis. (**D**) Treatment with HJP272 after AM had a minimal effect on fibrosis.

**Figure 7 ijms-25-04184-f007:**
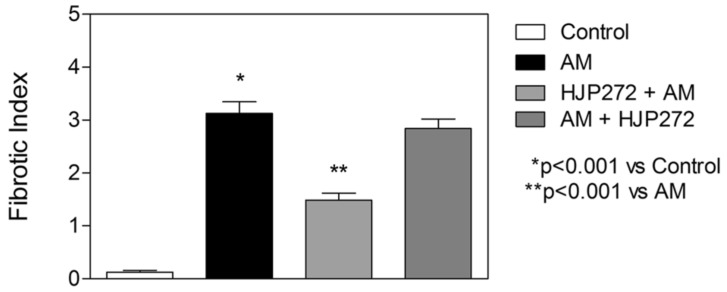
Graph showing the level of AM-induced fibrosis in the various treatment groups, as measured by the fibrotic index.

**Figure 8 ijms-25-04184-f008:**
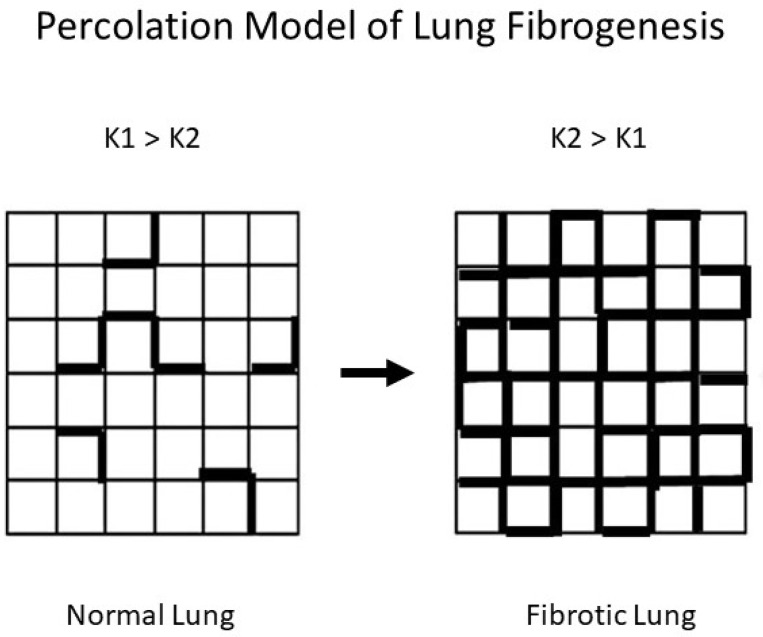
Fibrogenesis is represented by an increase in K2 (bold) units, which is associated with the redistribution of mechanical forces in the lung.

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
