# Peer review of "Maximizing the Therapeutic Effect of Endothelin Receptor Antagonists in Pulmonary Fibrosis: A Paradigm for Treating the Disease"

_ijms, 2024, doi:10.3390/ijms25084184_

Round 1
Reviewer 1 Report
Comments and Suggestions for Authors
The perspective article by Jerome Cantor is a very good attempt to provide a comprehensive overview on the therapeutic effect of Endothelin Receptor Antagonist (ERA) in pulmonary fibrosis (PF). Author stress the prophylactic advantage of ERA over treatment. I have below comments to improve the quality of the review.
1. Specify what is ERA in the abstract.
2. Make an illustration on how ET-1 involves in neutrophil recruitment, induction of CXCR2 expression in neutrophil, increased expression of macrophage TNFR1 and how HJP272 induce anti-inflammatory effect targeting ET-1 receptor. In summary, complete overview on ET-1 in pulmonary fibrosis.
3. For histopathology figures, provide color figures. Not grey scale. Also provide histology score as bar graph.
4. Provide more information on HJP272. Is it a drug in development stage, clinical trial or market?
5. It is confusing as author mixing ERAs and HJP272. My understanding is that HJP272 is one of the ERA? Or it is the only ERA. If many ERA available, how HJP272 is different?
6. The key perspective of this article is that ERA works as prophylaxis and not as a treatment. How author see a translational aspect of this key information. For example, do author suggest that individuals with occupational risk or familial risk for pulmonary fibrosis must take ERA as prophylaxis? Although author logic on biomarker based early treatment is very valid, will the risk population based treatment is useful?
7. Line No. 121, provide clinical trial information/reference to support that statement that ERA is ineffective in human pulmonary fibrosis.
8. Please expand this argument in Line No. 149 “the course of injury and repair is programmed at a very early stage of the inflammatory process”. Do author mean, if pulmonary fibrosis not treated early it cannot be treated forever?
9. Expand the discussion on influx of fibrocytes into the lung during PF (Line No. 167). How ERA block this phenomenon? If possible, make an illustration on this.
10. Ratio on free-DID vs. bound-DID needs more support. Author referred an unpublished data of a clinical trial. How big the difference is? Many biomarkers are failing badly in clinical trial as the difference between two set-up is marginal. The confounding factors such as age, sex, ethnicity, disease stage etc., needs to be considered (author himself accounted the fact in section 4.3).
11. Fig.5 Model is very interesting. Can author get help from people in mathematical modeling to improve it better?
12. Reduce the words in conclusion section. Write only the key perspectives.
13. Author discuss majorly his own studies. I suggest to include further clinical and preclinical statements and references to support his perspective.
Comments on the Quality of English LanguageGood.
Author Response
- We have specified what ERA denotes in the abstract (lines 10-12).
- An illustration of the mechanism of AM-induced pulmonary fibrosis and the stages where ET-1 plays a role is included in the revised manuscript (new figure 5).
- We have provided available color figures (2 and 6) and included graphs of the fibrosis index for each treatment group.
- We indicate that HJP272 has not been developed for commercial use (lines 49-51).
- On lines 177-182, we describe the different features of HJP272 and other ERAs (lines 177-182).
- On lines 132-133, we indicate that ERAs may be used as prophylactic agents to prevent the lung fibrosis side-effects of BLM, AM, and other drugs. Most cases of pulmonary fibrosis arise spontaneously and therefore cannot be treated prophylactically.
- We have provided multiple references indicating the limited efficacy of ERAs in pulmonary fibrosis (lines 185-187).
- On lines 213-218, we specify that once the disease undergoes a phase transition it is less amenable to treatment, not untreatable.
- We have added a sentence more fully explaining the role of ET-1 on fibrocyte differentiation (lines 175-176).
- We provide additional referenced support for the use of the free to peptide-bound DID ratio. We also provide more information about the unpublished results, including the p-value (lines 239-243).
- We did not have the opportunity to revise the original figure 5.
- We reduced the size of the Conclusion to emphasize only the critical arguments in the paper.
- Several additional references were added to the paper regarding the role ERAs in BLM-induced pulmonary fibrosis (refs 13, 14).
Reviewer 2 Report
Comments and Suggestions for Authors
The topic is quite interesting but it is not easy to understand the type of paper. I think it is a sort of narrative review but the title is not very informative. The section should be clearly introduced in the introduction (the aim is ...; we are going to present the following issues: etc etc). So I suggest to re-organiza the paper in order to make clear to the reader the aim and the thread.
Author Response
We have included a new opening paragraph that provides the aims and main arguments of the paper.
Reviewer 3 Report
Comments and Suggestions for Authors
In this article, the author discusses the potential of endothelin receptor antagonists (ERA) as a therapeutic agent for pulmonary fibrosis. The author bases his concept on his own preclinical studies, in which on animal models (with the administration of bleomycin and amiodarone) he demonstrated the positive antifibrotic effect of ERA when administered in the early phase. This part is well argued.
Further, the author rightly notes that with the use of biomarkers, ERA therapy may be more successful. But, however, here follows the author’s proposal, which is not supported by any studies with ERA - to use the ratio of free to peptide-bound desmosine as a biomarker, which already looks like a “pure theory”. So, stronger arguments are needed.
Author Response
We have added a statement regarding the actual correlation between an increase in free to peptide-bound DID and loss of lung function in human pulmonary emphysema. We also indicate that the use of this putative biomarker to detect the effects of therapeutic intervention is speculative (lines 242-247).
Round 2
Reviewer 2 Report
Comments and Suggestions for Authors
None of my comments were addressed.
Reviewer 3 Report
Comments and Suggestions for Authors
The authors took into account the comments and concerns of the reviewers and made some additions to the paper. In this form, the article can be accepted for publication.